# Fall Prediction Based on Instrumented Measures of Gait and Turning in Daily Life in People with Multiple Sclerosis

**DOI:** 10.3390/s22165940

**Published:** 2022-08-09

**Authors:** Ishu Arpan, Vrutangkumar V. Shah, James McNames, Graham Harker, Patricia Carlson-Kuhta, Rebecca Spain, Mahmoud El-Gohary, Martina Mancini, Fay B. Horak

**Affiliations:** 1Department of Neurology, Oregon Health & Science University, Portland, OR 97239, USA; 2Advanced Imaging Research Center, Oregon Health & Science University Portland, OR 97239, USA; 3APDM Wearable Technologies-A Clario Company, 2828 S Corbett Ave, Ste 135, Portland, OR 97201, USA; 4Department of Electrical and Computer Engineering, Portland State University, 1825 SW Broadway, Portland, OR 97201, USA

**Keywords:** home monitoring, instrumented gait and turning analysis, retrospective fall history, prospective falls, multiple sclerosis, pitch at toe-off

## Abstract

This study investigates the potential of passive monitoring of gait and turning in daily life in people with multiple sclerosis (PwMS) to identify those at future risk of falls. Seven days of passive monitoring of gait and turning were carried out in a pilot study of 26 PwMS in home settings using wearable inertial sensors. The retrospective fall history was collected at the baseline. After gait and turning data collection in daily life, PwMS were followed biweekly for a year and were classified as fallers if they experienced >1 fall. The ability of short-term passive monitoring of gait and turning, as well as retrospective fall history to predict future falls were compared using receiver operator curves and regression analysis. The history of retrospective falls was not identified as a significant predictor of future falls in this cohort (AUC = 0.62, *p* = 0.32). Among quantitative monitoring measures of gait and turning, the pitch at toe-off was the best predictor of falls (AUC = 0.86, *p* < 0.01). Fallers had a smaller pitch of their feet at toe-off, reflecting less plantarflexion during the push-off phase of walking, which can impact forward propulsion and swing initiation and can result in poor foot clearance and an increased metabolic cost of walking. In conclusion, our cohort of PwMS showed that objective monitoring of gait and turning in daily life can identify those at future risk of falls, and the pitch at toe-off was the single most influential predictor of future falls. Therefore, interventions aimed at improving the strength of plantarflexion muscles, range of motion, and increased proprioceptive input may benefit PwMS at future fall risk.

## 1. Introduction

Multiple sclerosis leads to the disruption of neurological networks in the central nervous system, commonly affecting functions such as mobility, muscle strength, and coordination [1,2]. Impairments in any of these functions can contribute to the common occurrence of falling. A meta-analysis found that the prevalence of falls is about 56% in people with MS (PwMS), and around 37% of fallers are classified as frequent fallers [3,4,5]. Falling is a significant concern for PwMS due to an increased chance of injury and reduced physical and professional activities and societal participation [3,6]. Therefore, falls in MS are gaining increased attention in the scientific community, driving research on multivariate risk prediction models using prospective study designs to accurately predict and prevent future occurrences.

Impaired balance and gait are the most common causes of falls in PwMS [7,8,9,10]. Studies have shown that PwMS are more prone to falls during dynamic activities such as walking and turning, rendering quantitative investigation of balance dysfunction during walking and turning critical to identify fall-prone individuals. Wearable inertial sensors are sensitive to early changes in balance control and can detect mobility dysfunction in PwMS earlier than conventional measures [11]. Furthermore, wearable technology can provide quantitative gait and turning assessments even in-home settings [12,13,14,15,16]. In PwMS, most falls are known to occur inside the home during general mobility without the execution of any other specific task. Hence, the investigation of gait and turning dysfunction in the home environment is crucial for our understanding of the environmental context of falls. Moreover, brief examinations of balance and gait in the clinic or research settings may not accurately reflect the actual functional mobility of patients in their everyday lives. Mobility can fluctuate due to many different factors, such as fatigue, medication, and environmental conditions, which are likely for a person with MS. Therefore, passive monitoring of mobility during daily life could help better assess the risk of falling in PwMS, allowing researchers and clinicians to gain insights about their patients both inside and outside of healthcare facilities.

While mobility dysfunction remains the center of investigation for future fall prediction and prevention studies, recent evidence suggests that clinical balance measures, in isolation, may be poor predictors of future fallers. A study by Cameron et al. advocates that a positive retrospective fall status of patients is the single most influential predictor of falls in PwMS [17]. Although this approach may seem simple, a downside of using fall history to predict fall risk is that it can only be implemented in those patients who have experienced at least one fall in the past, limiting its applicability in identifying those at risk of the first fall episode. Furthermore, the previous study only focused on clinical measures of disease severity, subjective balance, gait questionnaires, and a handful of objective clinical gait characteristics such as gait speed as predictors of future fallers in MS [17]. Therefore, multifactorial assessment including both fall history and quantitative measures of gait and turning in daily life may provide a better fall risk screening tool.

The objectives of this study were (1) to identify whether short-term passive monitoring of gait and turning (mobility) measures in daily life alone or in combination with a history of falls had the highest discriminative ability for predicting future fallers from non-fallers in PwMS, and (2) to assess the potential value of a multivariate prediction model that combined both subjective fall history and gait and turning measures in daily life in identifying those at risk of future falls.

## 2. Methods

This prospective cohort study was approved by the Institutional Review Board of Oregon Health & Science University. PwMS were recruited from the outpatient MS specialty clinics of the institution and surrounding community neurology clinics. The investigation was conducted according to the principles expressed in the Declaration of Helsinki, and written informed consent was obtained from the participants.

Inclusion criteria for the participants included: (1) confirmed diagnosis of relapsing-remitting or progressive MS, (2) mild-to-moderate MS-associated disability, (3) complaints about mobility, (4) capable of communicating with investigators and able to follow instructions, (5) able to walk independently without an assistive device, and (6) no other neurological or musculoskeletal disorder that could affect mobility other than MS.

Passive monitoring of gait and turning during daily life were assessed using the Opal instrumented socks (detailed description provided elsewhere [18]). This system enables continuous characterization of the quantity and quality of gait and turning during daily life activities using one Opal sensor on the waist and wireless inertial sensors embedded in the socks (prototype instrumented socks; APDM Wearable Technologies-A Clario Company, Portland, OR, USA; Figure 1). Each Opal sensor includes a tri-axial accelerometer, gyroscope, and magnetometer with a sampling rate of 128 Hz. The Opal is lightweight (22 g), has a battery life of 12 h, and includes 8 GB of storage and can record over 30 days of data.

During a baseline visit to the clinic, participants learned how to wear the instrumented socks on each foot and one Opal sensor over the lower lumbar area with an elastic belt for continuous monitoring of mobility. They were instructed to wear the sensors for at least 8 h/day and then take them off to charge each night for a week. Raw data were stored in the 8 GB internal memory of the sensors and uploaded to a secure cloud-based database server for analysis after being mailed back to investigators.

**Gait and turning measures during daily life:** The algorithms used for extracting spatial and temporal measures of gait and turning have been detailed previously [18]. Briefly, the daily life algorithm first searches for possible walking bouts and turns from inertial sensor data of the feet and lumbar using a time-domain approach. Potential walking bouts are defined as at least 3 consecutive steps, at least 3 s in duration, and the duration from one step to the next step is less than 2.5 s. Finally, each potential bout is processed with the commercial gait analysis algorithms included in Mobility Lab (APDM Wearable Technologies-A Clario Company [18,19]). To precisely estimate the orientation and position trajectory of each foot between quiet stance periods, we fused the information from the accelerometers and gyroscopes using the unscented Kalman filter. For the results reported in this paper, we only included stride pairs during periods of straight walking, and we excluded walking during turns. To detect and characterize each turn, we used the algorithm described in El-Gohary et al. 2013 [20]. Overall, 35 digital outcome measures of mobility were obtained as described previously [21]. Specifically, we had 9 measures of mobility for lower body, 4 instrumented measures for turning, and 3 measures of trunk (Appendix A). In addition to quantity measures of mobility, we had 3 measures of quality of gait characteristics in daily life. We also evaluated the variability of each measure from all the gait strides and turns (16 total variability measures) across the 7 days as the coefficient of variation (CV) (standard deviation divided by the mean). The detailed description of the definition of mobility measure is given in Appendix A. For example, the pitch at toe-off is the plantar flexion angle of the foot relative to a level, horizontal surface at the time the foot leaves the floor at push-off during straight-ahead walking. With Opal sensors attached on top of the foot, the algorithms utilize the angular velocity of the foot to determine the change in angle from the time when the foot is flat to time the foot leaves the floor at push-off. The angular change from foot flat to time of toe-off, along with a kinematic model of the foot were used to estimate the pitch at toe-off.

**Fall monitoring (retrospective and prospective):** Participants were followed prospectively for a year for falls. Participants were instructed to make note of any falls and report the information during biweekly email surveys. A research assistant contacted participants (1) in cases of reported falls to find out the details or (2) if biweekly fall reports were not received. A fall was defined as “an event that results in coming to rest unintentionally on the ground or other lower-level” [22,23]. Subjects were classified as fallers if they had more than >1 fall in the 12-month period after home monitoring.

**Statistical analysis:** The Shapiro–Wilk test was used to test the normality of the data. Independent t-tests (or Mann–Whitney U-tests if not normally distributed) were used to compare between-group differences in fallers and non-fallers. Effect size was calculated using Cohen’s d.

Fall prediction based on instrumented measures of mobility (Univariate Model): The area under the receiver operating characteristic (ROC) curve (AUC) was computed for each gait measure that discriminated fallers from non-fallers and ordered the measures from the highest to the lowest AUC value.

Fall prediction based on the history of falls (univariate model): The relationship between the history of falls and prospective falls was investigated using logistic regression analysis, and the AUC was computed.

Final prediction model based on fall history and daily mobility measures (multivariate model): Regression analysis was performed to identify the best prediction model for falls. The history of falls along with instrumented measures of mobility were used to generate the risk model for prospective falls.

## 3. Results

### 3.1. Demographics

Table 1 shows the demographic characteristics of non-fallers (*n* = 13; 11 females (F) and 2 males (M)) and fallers (*n* = 13; 10F and 3M). No significant differences between the non-faller and faller groups were observed in age, weight, height, disease duration, or Expanded Disability Status Scale (EDSS) (Table 1). Daily life mobility data were collected on average for 6 days (range: 2–8 days) for an average total duration of 52 h (range: 17–78 h). Notably, no differences were observed between the quantity of mobility measures among fallers and non-fallers (Table 1); only the quality of mobility measures discriminated PwMS at future fall risk from non-fallers, as described below.

### 3.2. Fall Prediction Based on Instrumented Measures of Mobility

On average, the fallers walked slower than non-fallers with smaller stride lengths and spent a significantly greater percentage of the gait cycle in double-limb support during walking (Table 2). Similarly, the percentage of swing phase during gait cycle was reduced in fallers (37%) compared to non-fallers (39%). In addition, fallers demonstrated a significantly smaller pitch angle (plantarflexion) of the foot at toe-off during walking and smaller turning angles compared to non-fallers (Table 2). The top instrumented measures of mobility discriminating fallers from non-fallers in PwMS were the pitch angle of the foot at toe-off, gait speed, stride length, swing (%), and double-support (%) (Table 3).

### 3.3. Fall Prediction Based on the History of Falls

Retrospective fall status was not a significant predictor of prospective fall status in PwMS (AUC = 0.62, *p* = 0.32). The proportion of prospective fallers who did not have a history of falls in the past year was 46% (Figure 2).

Final prediction model based on fall history and daily mobility measures (multivariate model): When the instrumented gait measures discriminative of fallers from non-fallers (pitch at toe-off (°), gait speed (m/s), stride length (m), double-support (%), swing (%), pitch at initial contact (°), turn angle (°)) were entered along with the history of falls in the prediction model, a forward regression yielded a significant model consisting of only one gait variable, foot pitch at toe-off (*p* < 0.01). The discriminative ability of the final model to classify future fallers from non-fallers with the pitch at toe-off angle as a predictor was 86% (AUC = 0.86, *p* < 0.002, Figure 3).

## 4. Discussion

The increasing use of wearable technologies in movement disorders is revolutionizing health care since it allows quantitative assessment of balance and gait disorders in supervised clinical, as well as unsupervised daily life environments [24,25]. To the best of our knowledge, this pilot study is the first longitudinal study investigating the future fall risk in PwMS based on instrumented gait and turning monitoring in daily life. Measures of both the quantity and quality of mobility in daily life were obtained; however, only quality of mobility discriminated fallers from non-fallers.

A key finding of the study was that the foot pitch at toe-off was the single most influential predictor of the future fall risk in our modest sample of PwMS. The pitch at toe-off is the angle of the foot as it leaves the floor at push-off during straight-ahead walking. In our study, fallers had a significantly smaller lower pitch at toe-off in comparison to non-fallers, reflecting reduced plantarflexion during the push-off phase of walking. In PwMS, the reduced plantarflexion can be attributed to either the weakness of the gastrocnemius–soleus muscles and/or decreased foot/ankle proprioceptive input during walking. Deficits in the ankle plantarflexion at the push-off phase of the gait cycle can reduce the forward propulsion of the body and swing initiation by the trailing leg, resulting in poor foot clearance during the swing phase and decreased walking speed [26,27,28]. Besides alterations in the gait cycle, there are both mechanical and energetic consequences of reduced ankle plantar flexion in human walking [27]. The decreased plantar flexion at toe-off can increase the mechanical loading borne by the leading leg at heel-strike and increase the metabolic cost of walking [26,27]. Therefore, fall prevention programs aimed at improving strength training of plantar flexor muscles, ankle range of motion, and increased proprioceptive feedback at the ankle may prove beneficial in improving the dynamic stability and metabolic cost of walking in PwMS.

In addition to the reduced pitch angle at toe-off, several alterations in the spatiotemporal features of gait were observed for PwMS at fall risk. Our results supported the evidence from previous studies that gait speed with a cut-off 1.0 m/s could represent a useful tool for identifying individuals who are at risk of falling [29]. At least 77% of fallers in this study had a gait speed of <1.0 m/s in their daily lives compared to only 31% of non-fallers. The reduced gait speed observed in fallers can be attributed to either plantar flexor deficits at toe-off (as discussed above) or to a cautious gait strategy adopted by fallers to maintain dynamic balance by walking slowly and taking shorter steps with more time in double-support [30,31,32].

The gait cycle can be divided into two primary phases: the stance and swing phases, which alternate for each lower limb. The initial contact of the foot with the ground marks the beginning of the stance phase. Fallers in this study demonstrated significantly lower pitch angles at the initial foot contact compared to non-fallers, reflecting reduced dorsiflexion at foot strike. This finding combined with the reduced PF during the push-off phase indicates that PwMS at fall-risk tend to shuffle [21,33] or drag their feet when they walk. In addition, we found that fallers spent a significantly greater percentage of the gait cycle in the stance phase, specifically double-limb support time. Since individuals at fall risk may have better control over their center of mass movement when both feet are in contact with the ground simultaneously, increasing the percentage of double-support period during walking may reflect a compensatory mechanism to stabilize the inefficient gait control. Importantly, similar alterations in gait patterns have been previously observed in elderly fallers versus non-fallers in clinical settings [31], supporting the idea that monitoring the deviations in these spatiotemporal variables of gait in daily life and/or clinical settings is crucial for distinguishing prospective fallers from non-fallers in community adults, as well as patient populations.

Besides alterations in gait patterns during straight walking, we also found significant differences in the turning angle among fallers and non-fallers in PwMS. Fallers had significantly smaller turning angles over the week of monitoring compared to non-fallers, indicating that fallers may avoid or find it difficult to control large turns. Alternatively, fallers may hesitate while turning such that hesitations >2 s would be counted as two, smaller turns. Overall, our findings demonstrate that only measures of the quality of gait were significant predictors of fall risk in PwMS, while the quantity of mobility was similar for fallers and non-fallers (Table 1).

Retrospective fall history was not a significant predictor of future falls in our cohort of PwMS, in contrast to the previous study [17]. This might be due to the small sample size of our study or the differences among methodologies to categorize fallers versus non-fallers across studies. We surveyed subjects via email every 2 weeks for their fall status, whereas most other studies had monthly fall diaries mailed, which increases the chance of missing falls.

The main limitation of our study is the small sample size, so it should be considered as pilot data for a larger study. However, the real-life mobility data collected from this study represent an important starting point to improve our knowledge on remote monitoring of gait in patients with MS. Second, we performed all mobility analyses by taking the mean of each measure for all the strides over a week for every participant and, thus, gave equal weight to each stride [21]. However, in reality, gait speed and other gait measures may vary among gait bouts of different lengths [16]. Hence, future studies are recommended to analyze the impact of bout length on each mobility measure and how gait bout length affects the discriminatory power of each mobility measure.

## 5. Conclusions

Our results demonstrated the potential of objective monitoring of gait and turning in daily life to identify those at the future risk of falls, even without a history of falls. Most of the gait impairments in fallers compared to non-fallers were consistent with a slower pace of gait as a decreased foot pitch angle at toe-off, slower gait speed, longer double-support time, and shorter stride length, and even smaller turn angles reflect a weaker or more cautious gait. Our finding of the decreased foot pitch angle at toe-off as a most critical predictor of falls may assist in future fall prevention by developing optimal interventions for this impairment, as well as by identifying PwMS in need of treatment to avoid falls.

## Figures and Tables

**Figure 1 sensors-22-05940-f001:**
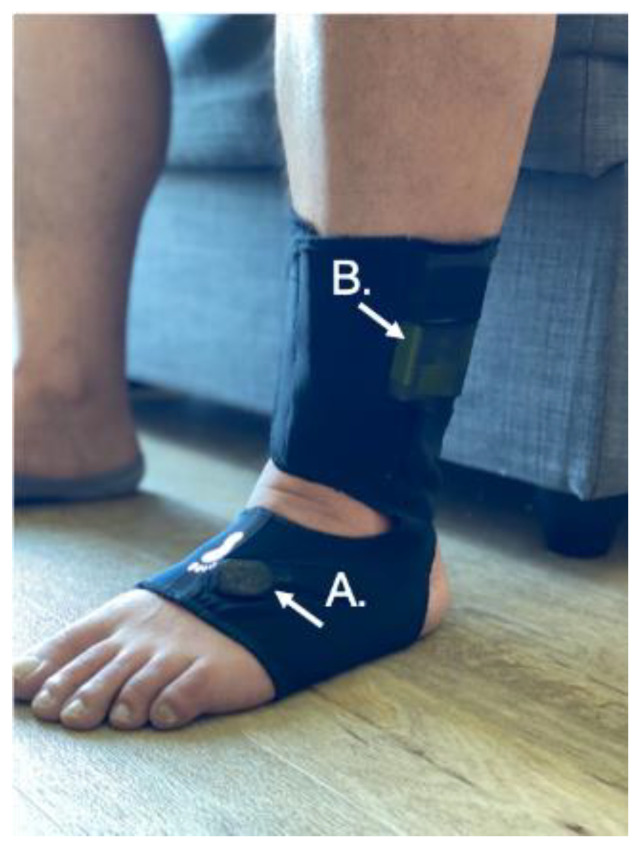
Participant wearing an instrumented sock, APDM prototype. The inertial sensor is located on top of the foot (**A**), and the main unit containing the battery in the socks is located in a second pocket just above the lateral malleolus (**B**). To maximize fit, the socks come in different sizes, and the Velcro attachment around the foot and ankle is adjustable to ensure a snug fit and that the sensor does not move on the foot while being worn.

**Figure 2 sensors-22-05940-f002:**
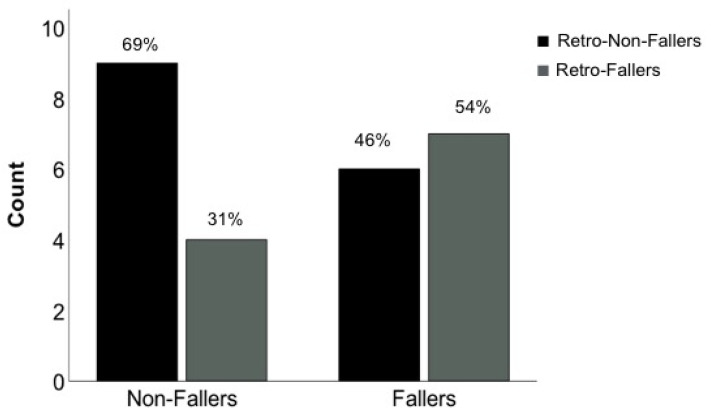
Percent of non-fallers and fallers predicted by retrospective fall history of 1 year.

**Figure 3 sensors-22-05940-f003:**
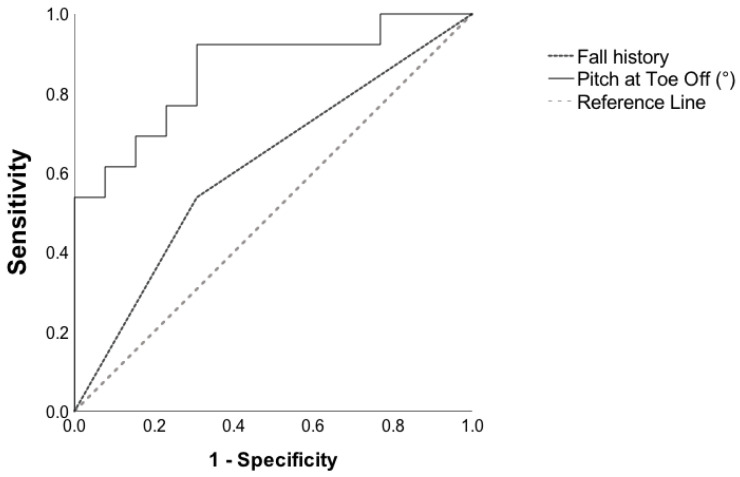
Receiver operating characteristic for fall history and pitch at toe-off angle to classify future fallers from non-fallers.

**Table 1 sensors-22-05940-t001:** Demographic features of the study population along with the quantity of mobility in daily life. Fallers were defined as people with MS who experienced more than 1 fall in the following year after their recruitment in the study.

DEMOGRAPHIC FEATURES		Faller/	N	Mean	Std. Error	*p* value
		Non-faller				
	Age (yrs)	Non-Fallers	13	49.2	2.4	0.1
		Fallers	13	49.1	3.5	
	EDSS (#)	Non-Fallers	13	4.3	0.23	0.8
		Fallers	13	4.2	0.18	
	Weight (lbs)	Non-Fallers	13	156.9	10.5	0.8
		Fallers	13	160.2	11.4	
	Height (cm)	Non-Fallers	13	170.2	2.2	1
		Fallers	13	170	3	
	Disease Duration (yrs)	Non-Fallers	13	13.8	2	0.4
		Fallers	13	16.8	2.9	
QUANTITY OF MOBILITY	Bouts/hour (#)	Non-Fallers	13	5.89	0.91	0.7
		Fallers	13	6.4	0.72	
	Strides/hour (#)	Non-Fallers	13	137.11	29	0.8
		Fallers	13	130.5	15.56	
	Turns/hour (#)	Non-Fallers	13	17.74	4.33	0.8
		Fallers	13	18.88	2.57	

**Table 2 sensors-22-05940-t002:** Differences among fallers and non-fallers in the instrumented gait and turning measures collected during the daily home monitoring.

Test Result Variable(s)		N	Mean	Std. Error	95% Confidence Interval		Range		*p* Value	Effect Size
					Lower	Upper	Min	Max		Cohen’s d
Pitch at Toe Off (°)	Non-Fallers	13	30.92	1.00	28.74	33.09	23.94	37.18	0.00	1.42
	Fallers	13	23.88	1.66	20.26	27.50	13.73	33.15		
Gait Speed (m/s)	Non-Fallers	13	1.08	0.03	1.01	1.16	0.90	1.26	0.01	1.05
	Fallers	13	0.89	0.06	0.76	1.03	0.40	1.31		
Stride Length (m)	Non-Fallers	13	1.22	0.03	1.15	1.30	1.00	1.41	0.01 ^a^	0.99
	Fallers	13	1.06	0.06	0.93	1.18	0.68	1.40		
Double Support (%)	Non-Fallers	13	22.70	0.70	21.17	24.23	18.48	26.94	0.01	1.14
	Fallers	13	26.14	0.95	24.06	28.22	21.07	31.29		
Swing (%)	Non-Fallers	13	38.68	0.35	37.91	39.44	36.58	40.76	0.01	1.13
	Fallers	13	37.03	0.45	36.05	38.02	34.47	39.46		
Pitch at Initial Contact (°)	Non-Fallers	13	22.09	1.09	19.73	24.46	26.51	12.46	0.02 ^a^	0.92
	Fallers	13	17.30	1.74	13.52	21.08	24.41	5.38		
Turn Angle (°)	Non-Fallers	12	88.79	1.40	85.71	91.87	79.03	95.58	0.04	0.87
	Fallers	13	82.55	2.43	77.25	87.86	63.36	97.65		

^a^ The data for stride length and pitch at initial contact were not normally distributed. Therefore, the Mann–Whitney U-test was used to compare between-group differences between fallers and non-fallers.

**Table 3 sensors-22-05940-t003:** The area under the receiver operating characteristic (AUC) curves to classify the instrumented measures of mobility as predictors of future falls.

Test Result Variable (s)	Area	Std. Error ^a^	Asymptotic Sig. ^b^	95% Confidence Interval	
				Lower	Upper
Pitch at Toe Off (°)	0.85	0.080	0.003	0.690	1.000
Gait Speed (m/s)	0.78	0.096	0.017	0.595	0.969
Stride Length (m)	0.78	0.100	0.019	0.579	0.972
Double Support (%)	0.78	0.095	0.017	0.589	0.962
Swing (%)	0.78	0.094	0.017	0.598	0.966
Pitch at Initial Contact (°)	0.77	0.093	0.020	0.587	0.951
Turn Angle (°)	0.75	0.104	0.034	0.546	0.954

^a^. Under the nonparametric assumption. ^b^. Null hypothesis: true area = 0.5.

## Data Availability

The data that support the findings of this study are available from the corresponding author upon request.

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
