# Peer review of "Fall Prediction Based on Instrumented Measures of Gait and Turning in Daily Life in People with Multiple Sclerosis"

_sensors, 2022, doi:10.3390/s22165940_

Round 1

Reviewer 1 Report

Thank you very much for the opportunity to review the proposed manuscript.  The investigators aimed to assess the potential of passive monitoring of gait and turning in daily life in people with multiple sclerosis to identify those at future risk of falls. 

The manuscript is clear, relevant for the field and presented in a well-structured manner. The cited references are mostly recent publications (within the last 5 years) and relevan.

However, I have a few critical concerns with the proposed work:

1.     How was the sample size estimated? 

2.     What was the effect size?

3.     Could you present the model of regression analysis (regression equation)? What steps were sequentially taken? How many variables were taken into model? 

There should be no more than 10-15 cases for one variable.

4.     The Author should discuss that the fall history is a poor indicator of the risk of falls. People who fall, don't often present an imbalance, it is just inattention, and those who do have a problem are highly cautious - so they don't fall.

5.  As the Discussion is long, the section Conclusion should be added and the most important finding highlighted.

Author Response

We are grateful to the reviewers for their time and constructive comments on our manuscript. We have implemented 1st reviewer’s comments and suggestions and wish to submit a revised version of the manuscript for further consideration in the journal. Changes in the initial version of the manuscript are either highlighted for the added sentences or strikethrough for the deleted sentences. Below, we provide a point-by-point response explaining how we have addressed the reviewer’s comments:

  1. How was the sample size estimated? 

The sample size was estimated based on a recent study on older adults [1]. We showed that variability of the number of steps during turning was a sensitive metric in predicting falls in the 6 months after the week of continuous monitoring in a group of healthy elderly fallers [1]. Out of 35 healthy elderly participants (sample of convenience), 7 fell at least once in the 6 months after the week of continuous monitoring. To determine the number of subjects needed in this study, we compared the variability of the number of steps needed to complete a turn by subjects who experienced one or more falls to variability in the subjects that did not fall. Given the fallers group mean variability of 0.59 (SD 0.04) and the non-fallers group mean of 0.54 (SD 0.03), for alpha = 0.05 and a power of 95%, we are adequately powered to separate fallers and non-fallers with a sample size of 12 subjects per group.

[1] Mancini, Martina, et al. "Continuous monitoring of turning mobility and its association to falls and cognitive function: a pilot study." Journals of Gerontology Series A: Biomedical Sciences and Medical Sciences 71.8 (2016): 1102-1108.

  1. What was the effect size?

 We have added effect sizes for instrumented gait measures discriminative of fallers from non-fallers in Table 2.

  1. Could you present the model of regression analysis (regression equation)? What steps were sequentially taken? How many variables were taken into model? 

Please see below for the details of the regression analysis in the paper:

Regression Model

Dependent variable: Prospective fall status

Independent variables: Pitch at Toe Off (°), Gait Speed (m/s), Stride Length (m), Double Support (%), Swing (%), Pitch at Initial Contact (°), Turn Angle (°), Fall History Status

 Regression: Forward regression using SPSS software (version 27).

For regression results, please see the attached document.

  1. The Author should discuss that fall history is a poor indicator of the risk of falls. People who fall, don't often present an imbalance, it is just inattention, and those who do have a problem are highly cautious - so they don't fall.

Although the reviewer is raising a good point, there is substantial evidence in the literature using the history of falls to predict future falls in both older adults [1]-[5] and people with MS [6]. In fact, it is widely known and accepted in the community that the Quality Standards Subcommittee of the American Academy of Neurology have recommended asking patients about their past falls to assess fall risk [7]. (Verbatim: “As for screening measures that may predict or further assess fall risk, a history of recent falls is an established predictor of future falls”)

[1]. Tinetti, Mary E., Mark Speechley, and Sandra F. Ginter. "Risk factors for falls among elderly persons living in the community." New England journal of medicine 319.26 (1988): 1701-1707.

[2]. Campbell, A. John, Michael J. Borrie, and George F. Spears. "Risk factors for falls in a community-based prospective study of people 70 years and older." Journal of gerontology 44.4 (1989): M112-M117.

[3]. Carpenter, Christopher R., et al. "Predicting geriatric falls following an episode of emergency department care: a systematic review." Academic Emergency Medicine 21.10 (2014): 1069-1082.

[4]. Jørgensen, Vivien, et al. "Falls and fear of falling predict future falls and related injuries in ambulatory." (2017).

[5]. Allen, Natalie E., Allison K. Schwarzel, and Colleen G. Canning. "Recurrent falls in Parkinson’s disease: a systematic review." Parkinson’s disease 2013 (2013).

[6]. Cameron, Michelle H., et al. "Predicting falls in people with multiple sclerosis: fall history is as accurate as more complex measures." Multiple sclerosis international 2013 (2013).

[7]. Thurman, David J., Judy A. Stevens, and Jaya K. Rao. "Practice parameter: assessing patients in a neurology practice for risk of falls (an evidence-based review): report of the Quality Standards Subcommittee of the American Academy of Neurology." Neurology 70.6 (2008): 473-479.

  1. As the Discussion is long, the section Conclusion should be added and the most important finding highlighted.

 Thank you for the comment. We have added the Conclusion section to the paper.

Reviewer 2 Report

Really nice piece of solid science. An important piece of research with a clear future potential for impact in people with MS. Well written and clear. 

given the low sample size the 0.05 cut-off to show results is pretty harsh. I would suggest to add all the measures and respective p-values in the supplementary.

It is known that many of the mobility measures change with the length of a bout as people start to 'stabilize' their walking behaviour.  The authors have shown that there is no general difference between groups in the quantity of mobility. Have the authors considered looking into how the gait measures change or not change with bout length and how this is predictive for fall risk? 

Author Response

Dear Reviewers and Editors,

Thank you for the careful consideration of our manuscript and the very useful recommendations and feedback. We have modified our manuscript in response to the reviewers’ concerns and in the attached cover letter, we will give a point-by-point reply to the reviewers’ recommendations. The changes in the revised manuscript are highlighted.

Thank you for a thorough review.

Reviewer 2

  1. Given the low sample size the 0.05 cut-off to show results is pretty harsh. I would suggest to add all the measures and respective p-values in the supplementary.

We have added all the instrumented gait and turning measures and their respective p-values in Table S2 supplementary section.

  1. It is known that many of the mobility measures change with the length of a bout as people start to 'stabilize' their walking behaviour.  The authors have shown that there is no general difference between groups in the quantity of mobility. Have the authors considered looking into how the gait measures change or not change with bout length and how this is predictive for fall risk? 

We agree with the reviewer. Our previous work has shown that various gait metrics are affected differently by bout length. We did not look at the effect of bout length in this project this was out of the scope of this manuscript. But given the importance of looking into the effect of bout length, we had already added this as a limitation of our work and we plan to consider it in the follow-up studies.

Reviewer 3 Report

The study is sound and interesting as it describes a long-term clinical analysis to determine the most adequate measure to predict the probability of suffering falls. I think this is not the most adequate journal for this type of studies as no contribution in the field of sensors, sensing technologies or sensing architectures is actually provided . Authors employ the measures provided by a commercial sensor. Nevertheless, I recommend to publish it because of their contributions to the predictions of fallers..

The methodology seems to be rigorous.

I suggest authors to address or discuss the following ideas:

The conclusion that ‘The history of retrospective falls was not identified as a significant 23 predictor of future falls’ is surprising and controversial. I suggest to review literature to contrast this with the conclusions of other clinical tests with fallers.

A more detailed description of the experimental testbed with the Opal sensors is required. Some operational aspects of the sensor should me mentioned: sampling rate, typology of the measured signals, sensor range, etc. Similarly, the algorithms used for extracting the measures of gait (in particular, pitch angle) should be at least briefly summarized.

Other aspects:

Lines 162-3: The reference for the well-known definition of fall (‘“an event that results 162 in coming to rest unintentionally on the ground or other lower-level”) is missing

The units are missing in some lines of table 1 (e.g. age, disease duration)

Some acronyms should be defined the first time they appear in the text. E.g. EDSS (Expanded Disability Status Scale?)

Do the conclusions change if the threshold to define a faller is modified (e.g. if fallers are defined as those suffering one or more falls during the observation period).

Comment if the number of subjects (26) and the observation period 81 year) correlates with those used in other similar studies.

Comment if the way users wear the socks may impact on the measures.

Minor aspects:

Is commented the total number of falls reported by the experimental users?

Table 2: The values of the lower and upper limits of the Pitch at Initial Contact are switched.

Table 2: The number of non-fallers for the measure of turn angle is just 12 (instead of 13). Is that a typo or just the measure is missing for one subject?

I recommend to include (if possible) a picture of the dataset (i.e a picture of a experimental user using the socks)

Author Response

Manuscript ID: sensors-1834541

Dear Reviewers and Editors,

Thank you for the careful consideration of our manuscript and the very useful recommendations and feedback. We have modified our manuscript in response to the reviewers’ concerns and in the attached cover letter, we will give a point-by-point reply to the reviewers’ recommendations. The changes in the revised manuscript are highlighted.

Thank you for a thorough review.

Reviewer 3

I suggest authors to address or discuss the following ideas:

  1. The conclusion that ‘The history of retrospective falls was not identified as a significant predictor of future falls’ is surprising and controversial. I suggest to review literature to contrast this with the conclusions of other clinical tests with fallers.

This is a good point. We modified the conclusion to the following: “The instrumented gait and turning measures from daily life may be more sensitive in predicting future falls in comparison to the fall history”.

  1. A more detailed description of the experimental testbed with the Opal sensors is required. Some operational aspects of the sensor should me mentioned: sampling rate, typology of the measured signals, sensor range, etc. Similarly, the algorithms used for extracting the measures of gait (in particular, pitch angle) should be at least briefly summarized.

Thank you. We have added the following information: “Each Opal sensor includes a tri-axial accelerometer, gyroscope, and magnetometer with a sampling rate of 128 Hz. The Opal is lightweight (22 g), has a battery life of 12 hours, and includes 8 GB of storage, that can record over 30 days of data.” For the algorithm used for extracting the gait measures, we had already provided the definition of how to calculate each of the gait measures in Supplementary Table S1. In addition, we added the following sentence for the pitch angle of the foot: “For example, the pitch at toe-off is the plantar flexion angle of the foot relative to a level, horizontal surface at the time the foot leaves the floor at push-off during straight-ahead walking. With Opal sensors attached on top of the foot, the algorithms utilize the angular velocity of the foot to determine the change in angle from the time when the foot was flat to the time the foot leaves the floor at push-off (see figure). The angular change from foot flat to time of toe-off, along with a kinematic model of the foot is used to estimate the pitch at toe-off.”

  1. Lines 162-3: The reference for the well-known definition of fall (‘“an event that results 162 in coming to rest unintentionally on the ground or other lower-level”) is missing

We have addressed this concern in the revised paper.

  1. The units are missing in some lines of table 1 (e.g. age, disease duration)

We have added units to all variables in Table 1

  1. Some acronyms should be defined the first time they appear in the text. E.g. EDSS (Expanded Disability Status Scale?)

We have addressed this concern in the revised paper.

  1. Do the conclusions change if the threshold to define a faller is modified (e.g. if fallers are defined as those suffering one or more falls during the observation period).

The reviewer has brought up a very interesting point. We expect the conclusion would not change much. But by changing the definition of faller as one or more falls, we are getting an imbalanced sample size: 5 non-fallers and 21 fallers, so we did not run the analysis.

  1. Comment if the number of subjects (26) and the observation period (1 year) correlates with those used in other similar studies.

The sample size of our study was relatively smaller (26 vs. >50 in most studies) compared to most studies in the literature that focused on predicting future falls in MS. However, an observation period of 6 months-1 year is pretty common for monitoring falls in this patient population. To address this concern, we have also added that this was a pilot study in the abstract now.

  1. Comment if the way users wear the socks may impact on the measures.

Thank you. As long as the sensor board is located somewhere on the top of the foot, the exact position and orientation of the sensor do not affect the accuracy of the algorithms. Further, the instrumented socks are adjustable with Velcro to ensure that a snug fit. We’ve added a more detailed description of how the sensor head is secured against the top of the foot by the sock in the revised manuscript and now reads as follows: “The main unit containing the battery is located in a second pocket just above the lateral malleolus. To maximize fit, the socks come in different sizes, and the Velcro attachment around the foot and ankle is adjustable to ensure that snug fit and that the sensor does not move on the foot while being worn.”

Minor aspects:

  1. Is commented the total number of falls reported by the experimental users?

Yes, the number of falls is reported biweekly by the users.

  1. Table 2: The values of the lower and upper limits of the Pitch at Initial Contact are switched.

Thank you for pointing it out. We have made corrections in the revised version.

  1. Table 2: The number of non-fallers for the measure of turn angle is just 12 (instead of 13). Is that a typo or just the measure is missing for one subject?

It is not typo. One participant in non-fallers group did not wear the lumbar sensor, and hence we are missing the lumbar sensor data.

  1. I recommend to include (if possible) a picture of the dataset (i.e a picture of a experimental user using the socks)

We have added a picture of a participant wearing the instrumented socks in Fig. 1 in the revised version.